# Polarization-Induced Phase Transitions in Ultra-Thin InGaN-Based Double Quantum Wells

**DOI:** 10.3390/nano12142418

**Published:** 2022-07-14

**Authors:** Sławomir P. Łepkowski, Abdur Rehman Anwar

**Affiliations:** Institute of High Pressure Physics—Unipress, Polish Academy of Sciences, ul. Sokołowska 29/37, 01-142 Warszawa, Poland; abdur12946@gmail.com

**Keywords:** topological phase transition, topological insulators, group-III nitrides, double quantum wells

## Abstract

We investigate the phase transitions and the properties of the topological insulator in InGaN/GaN and InN/InGaN double quantum wells grown along the [0001] direction. We apply a realistic model based on the nonlinear theory of elasticity and piezoelectricity and the eight-band **k·p** method with relativistic and nonrelativistic linear-wave-vector terms. In this approach, the effective spin–orbit interaction in InN is negative, which represents the worst-case scenario for obtaining the topological insulator in InGaN-based structures. Despite this rigorous assumption, we demonstrate that the topological insulator can occur in InGaN/GaN and InN/InGaN double quantum wells when the widths of individual quantum wells are two and three monolayers (*MLs*), and three and three *MLs*. In these structures, when the interwell barrier is sufficiently thin, we can observe the topological phase transition from the normal insulator to the topological insulator via the Weyl semimetal, and the nontopological phase transition from the topological insulator to the nonlocal topological semimetal. We find that in InGaN/GaN double quantum wells, the bulk energy gap in the topological insulator phase is much smaller for the structures with both quantum well widths of 3 *MLs* than in the case when the quantum well widths are two and three *MLs*, whereas in InN/InGaN double quantum wells, the opposite is true. In InN/InGaN structures with both quantum wells being three *MLs* and a two *ML* interwell barrier, the bulk energy gap for the topological insulator can reach about 1.2 meV. We also show that the topological insulator phase rapidly deteriorates with increasing width of the interwell barrier due to a decrease in the bulk energy gap and reduction in the window of In content between the normal insulator and the nonlocal topological semimetal. For InN/InGaN double quantum wells with the width of the interwell barrier above five or six *MLs*, the topological insulator phase does not appear. In these structures, we find two novel phase transitions, namely the nontopological phase transition from the normal insulator to the nonlocal normal semimetal and the topological phase transition from the nonlocal normal semimetal to the nonlocal topological semimetal via the buried Weyl semimetal. These results can guide future investigations towards achieving a topological insulator in InGaN-based nanostructures.

## 1. Introduction

Topological insulators (TIs) are a new class of materials that are characterized by an energy gap in the bulk electronic band structure and metallic states at the boundaries [1]. Closing of the band gap by the surface or edge states is caused by the nontrivial topology of the bulk states, originating from an inversion in the order of the valence and conduction bands at time-reversal-invariant wave vectors in the Brillouin zone [2]. This band inversion changes the Z_2_ topological invariant and causes the topological phase transition (TPT) between the normal insulator (NI) and the TI [2]. In 2D TIs, the band structure corresponds to the quantum spin Hall effect (QSHE), in which 1D gapless edge states are inside the bulk 2D sub-band spectrum [1,2]. The QSHE has been realized in topological 2D crystals and nanostructures [3,4,5,6,7,8,9]. In the case of 2D nanostructures, the TI phase has been only experimentally confirmed in two quantum well (QW) material systems, namely in HgTe/CdTe and InAs/GaSb/AlSb QWs [7,9]. In HgTe/CdTe QWs, the TI occurs due to the inverted band structure of HgTe, caused by the strong spin–orbit interaction (SOI), which leads to the inversion of the lowest conduction sub-band (CB) and the highest heavy-hole sub-band (HH) in structures with the QW width above a critical value of 6.4 nm [6,7]. Increasing the QW width above 12 nm leads to the nontopological phase transition (NTPT) from the TI to the nonlocal topological semimetal (NTSM), which arises from nonlocal overlapping between the sub-bands [10,11]. The TI phase in InAs/GaSb/AlSb QWs originates from the fact that the valence band (VB) of GaSb is 150 meV higher than the CB in InAs and the TPT can be achieved by varying the widths of the InAs and GaSb layers [8,9].

The 2D TIs were also proposed in InN/GaN QWs grown along the [0001] direction, parallel to the *c* axis of the wurtzite lattice [12]. In these structures, the extremely large built-in electric field originating from the piezoelectric effect and spontaneous polarization may invert the ordering of the CB and VB sub-bands according to the quantum confined Stark effect (QCSE), leading to a polarization-driven TPT [12]. The huge built-in electric field also induces the Rashba SOI, which significantly influences the bulk energy gap in the TI phase, $E_{2Dg}^{TI}$. Although GaN and InN are technologically important semiconductors, the issue of the SOI in these materials is still under scientific debate [13]. In InN/GaN topological QWs, the $E_{2Dg}^{TI}$ can reach 5 meV when the positive SOI of the order of a few milli-electron volts is assumed in GaN and InN crystals, or it can be about 1.25 meV when the negative SOI in InN is considered [12,13]. Although these values of $E_{2Dg}^{TI}\;$are significantly smaller than that for HgTe/CdTe and InAs/GaSb/AlSb QWs [14,15,16], they are large enough to allow for the experimental verification of the QSHE in these structures [17,18]. Unfortunately, the problem is that to achieve the TPT in InN/GaN QWs, the QW width should be at least four monolayers (*MLs*) and the growth of such thick and fully strained structures is extremely difficult, due to large strain reaching 11%. This problem can be partially overcome by applying InGaN alloys [13,19,20]. However, in In_x_Ga_1−x_N/GaN QWs, the critical thickness for obtaining the TI state increases faster with decreasing In content in the QWs than the critical thickness for pseudomorphic growth [13,20]. The situation is more promising in InN/In_y_Ga_1−y_N QWs, where the critical thickness for obtaining the TI state increases slower with increasing In content in the barriers than the critical thickness for the pseudomorphic growth [13]. It was also predicted that in InN/In_y_Ga_1−y_N QWs with a barrier In content of less than 0.5, the $E_{2Dg}^{TI}$ is about 2 meV, assuming a negative SOI in InN [13]. Despite multiple attempts, the growth of topological InGaN-based QWs remains a challenge [21,22,23]. Further research toward obtaining the TI state in group-III nitride nanostructures is desirable due to numerous future applications of these nanomaterials in electronics, piezotronics, spintronics, and quantum computing [24,25,26,27].

The investigations of the TPT in 2D semiconductor systems have recently been extended to double QWs (DQWs). In these structures, the TPT depends not only on the thickness of the individual QWs, but also on the width of the interwell barrier,$\;L_{ib}$, which determines the tunnel-induced hybridization between the QW subbands. In particular, it was shown that in tunnel-coupled HgTe/CdTe DQWs, the TI phase can be achieved when the thickness of the individual QWs is significantly smaller than the critical thickness for obtaining the TI state in single QW structures [28,29]. The picture of phases in HgTe/CdTe DQWs is richer than in the single QWs. In symmetric HgTe-based DQWs with an inverted ordering of sub-bands, apart from the NI, TI, and NTSM phases, there is a semimetal phase that holds similar properties to bilayer graphene (BG) [29]. This BG phase was experimentally confirmed by local and nonlocal resistance measurements [30].

In this work, we investigated the phase transitions and the properties of the TI phase in In_x_Ga_1−x_N/GaN and InN/In_y_Ga_1−y_N DQWs grown along the [0001] direction (see Figure 1). We applied a model based on the nonlinear theory of elasticity and piezoelectricity and the eight-band **k·p** method with relativistic and nonrelativistic linear-wave-vector terms. In our approach, the effective SOI in InN is negative [31], which represents the worst-case scenario for obtaining the TI in InGaN-based structures [13]. Despite this rigorous assumption, we demonstrate that the TI phase can occur in In_x_Ga_1−x_N/GaN and InN/In_y_Ga_1−y_N DQWs when the widths of the individual QWs are two and three *MLs*, and three and three *MLs*. In these structures, when the interwell barrier is sufficiently thin, one can achieve the TPT from the NI to the TI and the NTPT from the TI to the NTSM. We found that in In_x_Ga_1−x_N/GaN DQWs, the $E_{2Dg}^{TI}$ is much smaller for the structures with both QW widths having three *MLs* than in the case when the QW widths are two and three *MLs*, whereas in InN/In_y_Ga_1−y_N DQWs, the opposite is true. For InN/In_y_Ga_1−y_N DQWs with both QWs having widths of three *MLs* and the $L_{ib}$ of two *MLs*, the $E_{2Dg}^{TI}$ can reach about 1.2 meV. Our calculations also revealed that the $E_{2Dg}^{TI}$ rapidly decreases with an increasing $L_{ib}$. We found that for InN/In_y_Ga_1−y_N DQWs with the $L_{ib}$ above five or six *MLs*, the TI phase does not appear, and two novel phase transitions occur.

## 2. Theoretical Model

To study the polarization-induced phase transitions in InGaN-based DQWs, we employed the 8-band **k∙p** method combined with the nonlinear theory of elasticity and piezoelectricity. The application of the nonlinear theory of elasticity and piezoelectricity enabled us to accurately describe strain, piezoelectric polarization, and the built-in electric field, which is essential for obtaining an inversion of the CB and VB sub-bands. The applied **k∙p** method calculates quantum states in InGaN-based DQWs on the assumption that the effective SOI in InN is negative, which is crucial for determining the nature of the TPT and the properties of the TI phase.

We consider the structures (see Figure 1) in which the chemical compositions of the substrate, external barriers, and interwell barrier are the same, so these layers are unstrained. For simplicity, we also assume that the chemical compositions of both QWs are identical. Strain is only present in the QWs and is described by the following tensor:(1)$$\varepsilon =\left[\begin{array}{ccc} \varepsilon_{xx} & 0 & 0\\ 0 & \varepsilon_{xx} & 0\\ 0 & 0 & \varepsilon_{zz}=-R_{B}\varepsilon_{xx} \end{array}\right]$$
where $\varepsilon_{xx}$ is the in-plane strain, $\varepsilon_{zz}$ is the out-of-plane strain, and $R_{B}$ denotes the biaxial relaxation coefficient [32]. The in-plane strain is determined by the well-known formula $\varepsilon_{xx}=\frac{a_{s}}{a_{qw}}-1$, where $a_{S}$ and $a_{qw}$ are the lattice constants of the substrate and the QW material, respectively. We took the a lattice constants for GaN and InN from [33] and assumed that for InGaN alloys, they linearly depend on composition [34]. The $R_{B}$ coefficient is usually determined using the linear theory of elasticity, which predicts that it is equal to $2\frac{C_{13}}{C_{33}}$, where $C_{13}$ and $C_{33}$ are the second-order elastic constants. This simple approach is, however, inaccurate when the strain $\varepsilon_{xx}\;$is large [32,35]. Here, we apply a more general formula for the $R_{B}$ coefficient, which we derived in the framework of the third-order elasticity theory, as follows,
(2)$$R_{B}=\frac{1}{\varepsilon_{xx}}\left[1-\sqrt{1+\frac{2}{C_{333}}\left(-c+\sqrt{c^{2}-2C_{333}d}\right)}\right]$$
where $c=C_{33}+2C_{133}\left(\varepsilon_{xx}+\frac{1}{2}\varepsilon_{xx}^{2}\right)$ and $d=2C_{13}\left(\varepsilon_{xx}+\frac{1}{2}\varepsilon_{xx}^{2}\right)+\left(C_{113}+C_{123}\right){\left(\varepsilon_{xx}+\frac{1}{2}\varepsilon_{xx}^{2}\right)}^{2}$ [24,32]. In the above formula, $C_{113}$, $C_{123}$, $C_{133}$, and $C_{333}$ are the third-order elastic constants. For GaN and InN, we use the values of the elastic constants obtained from ab initio calculations, which were performed using the relationship between strain and the Helmholtz free energy density [32]. For InGaN alloys, we consider the nonlinear composition dependencies of the second-order elastic constants [35,36]. The composition dependencies of the third-order elastic constants are unknown for the group-III nitride alloys and, therefore, we use the linear approximation for these parameters in InGaN.

The built-in electric field in DQWs is calculated using a simple analytic model derived for a multilayer structure in [37]. This model is based on the assumption that the potential drop over the entire DQW structure, consisting of two external barriers, two QWs, and the interwell barrier, vanishes. The values of the built-in electric field in the corresponding layers of the DQW structure are given by the following formula:(3)$$E_{i}=\frac{\sum_{k=1}^{5}\left[\frac{L_{k}}{\lambda_{k}}\left(P_{k}-P_{i}\right)\right]}{\lambda_{i}\sum_{k=1}^{5}\left(\frac{L_{k}}{\lambda_{k}}\right)},\;\;\;\;\;\;i=1,\ldots ,5$$
where $L_{i}$, $P_{i}$, and $\lambda_{i}$ denote the width of a layer, macroscopic polarization, and electric permittivity, respectively [37]. In this work, we dealt with the DQWs consisting of ultra-thin QWs and an interwell barrier, with widths expressed in *MLs*. The well widths depend on strain as follows,
(4)$$L_{qw}=\frac{1}{2}n_{qw}c_{qw}\left(1-R_{B}\varepsilon_{xx}\right)$$
where $n_{qw}$ is the number of *MLs*, and $c_{qw}$ denotes the c lattice constant of the QW material. The factor of $\frac{1}{2}$ in Formula (4) originates from the fact that the wurtzite unit cell contains two *MLs*. We take the c lattice constants for GaN and InN from [33] and assume that for InGaN alloys, they linearly depend on composition [34]. In QWs, the macroscopic polarization is the sum of the spontaneous polarization $P_{sp}$ and the piezoelectric polarization $P_{pz}$, so it can be expressed by
(5)$$P=P_{sp}+P_{pz}=P_{sp}+2e_{31}\varepsilon_{xx}+e_{33}\varepsilon_{zz}+\left(B_{311}+B_{312}\right)\varepsilon_{xx}^{2}+\frac{1}{2}B_{333}\varepsilon_{zz}^{2}+2B_{313}\varepsilon_{xx}\varepsilon_{zz}$$
where $e_{31}$ and $e_{33}$ are the first-order piezoelectric constants; $B_{311}$, $B_{312}$, $B_{333}$, and $B_{313}$ are the second-order piezoelectric constants [38]. For unstrained barriers, the piezoelectric polarization is zero, and we have $P=P_{sp}$.

The electronic states in InGaN-based DQWs are calculated using the 8-band **k·p** Hamiltonian $H^{8\times 8}$ with relativistic and nonrelativistic linear-wave-vector terms, which were parametrized according to ab initio calculations performed using the quasiparticle self-consistent GW method [13,31]. The Hamiltonian $H^{8\times 8}$ is represented in a matrix form as follows:(6)$$H^{8\times 8}=\left[\begin{array}{cccccccc} H_{c} & -Q & Q^{*} & R & 0 & 0 & 0 & 0\\ -Q^{*} & F & K^{*} & M_{-}^{*} & 0 & 0 & -W^{*} & 0\\ Q & K & G & -N_{+} & 0 & -W^{*} & -T & \sqrt{2}\Delta_{3}\\ R & M_{-} & -N_{+}^{*} & L & 0 & 0 & \sqrt{2}\Delta_{3} & -S^{*}\\ 0 & 0 & 0 & 0 & H_{c} & Q^{*} & -Q & R\\ 0 & 0 & -W & 0 & Q & F & K & -M_{+}\\ 0 & -W & -T^{*} & \sqrt{2}\Delta_{3} & -Q^{*} & K^{*} & G & N_{-}^{*}\\ 0 & 0 & \sqrt{2}\Delta_{3} & -S & R & -M_{+}^{*} & N_{-} & L \end{array}\right]\begin{array}{c} |iS,↑\rangle \\ |-\left(X+iY\right)/\sqrt{2},↑\rangle \\ |\left(X-iY\right)/\sqrt{2},↑\rangle \\ |Z,↑\rangle \\ |iS,↓\rangle \\ |\left(X-iY\right)/\sqrt{2},↓\rangle \\ |-\left(X+iY\right)/\sqrt{2},↓\rangle \\ |Z,↓\rangle \end{array}$$
where $H_{c}=E_{vb}+E_{g}+A_{c⟘}k_{⟘}^{2}+A_{c||}k_{z}^{2}$, $Q=P_{2}k_{+}/\sqrt{2}$, $R=P_{1}k_{z}$, $F=\Delta_{1}+\Delta_{2}+\left(A_{2}+A_{4}\right)k_{⟘}^{2}+\left(A_{1}+A_{3}\right)k_{z}^{2}$, $G=F-2\Delta_{2}$, $L=A_{2}k_{⟘}^{2}+A_{1}k_{z}^{2}$, $K=A_{5}k_{+}^{2}$, $M_{+}=\left[A_{6}k_{z}+i\left(A_{7}+\alpha_{4}\right)\right]k_{+}$, $M_{-}=\left[A_{6}k_{z}-i\left(A_{7}+\alpha_{4}\right)\right]k_{+}$, $N_{+}=\left[A_{6}k_{z}+i\left(A_{7}-\alpha_{4}\right)\right]k_{+}$, $N_{-}=\left[A_{6}k_{z}-i\left(A_{7}-\alpha_{4}\right)\right]k_{+}$, $S=i\alpha_{1}k_{+}$, $T=i\alpha_{2}k_{+}$, and $W=i\left(\alpha_{1}+\alpha_{3}\right)k_{+}$. The top valence band energy and energy gap are denoted by $E_{vb}$ and $E_{g}$, respectively; $A_{c⟘}$ and $A_{c||}$ describe the dispersion of the CB; whereas $P_{1}$ and $P_{2}$ are the Kane parameters [13,20]. The valence band parameters $A_{1},\ldots ,A_{7}$, $\alpha_{1},\ldots ,\alpha_{4}$, and $\Delta_{1},\ldots ,\Delta_{3}$ were taken from [31] for GaN and InN, whereas for InGaN alloys, the linear approximation was applied. Additionally, the parameters $A_{1},\ldots ,A_{6}$ were rescaled according to [20]. Strain and the built-in electric field were included in the Hamiltonian $H^{8\times 8}$ according to [39,40]. Then, replacing $k_{z}$ in the Hamiltonian $H^{8\times 8}$ by the operator $-i\frac{\partial }{\partial z}$, we have the 8-band Schrödinger-type equation,
(7)$$\overset{8}{\underset{\beta =1}{\sum }}H_{\alpha ,\beta }^{8\times 8}\left({\vec{k}}_{⟘},\;k_{z}=-i\frac{\partial }{\partial z}\right)F_{m,\beta }\left(z,{\vec{k}}_{⟘}\right)=E_{m}\left({\vec{k}}_{⟘}\right)F_{m,\alpha }\left(z,{\vec{k}}_{⟘}\right),\;\;\;\;\;\alpha =1,\ldots ,8,$$
where $E_{m}\left({\vec{k}}_{⟘}\right)$ and $F_{m,\beta }\left(z,{\vec{k}}_{⟘}\right)$ are the energies and the envelope functions of the DQW states, respectively [13,20]. Because the material parameters depend on position in DQW structures, we use the standard symmetrization to ensure the Hermiticity of operators containing the products of functions and derivatives [20]. Equation (7) is solved using the standard finite element method [41].

## 3. Results and Discussion

We considered In_x_Ga_1−x_N/GaN and InN/In_y_Ga_1−y_N DQWs with the widths of individual QWs, $L_{qw,A}$, and $L_{qw,B}$, equal to two and three *MLs*, three and two *MLs*, and three and three *MLs*. We found that TPT can occur in these structures. On the other hand, in thinner DQWs, i.e., when $L_{qw,A}=2\;MLs$ and $L_{qw,B}=2\;MLs$, the QCSE is too weak to induce the TPT, and only the NI phase appears. We assumed that the thickness of the external barriers (see Figure 1) is large, i.e., $L_{eb}=2000\;\mathrm{nm}$, because this makes the built-in electric field in QWs extremely large and the TPT easier to achieve [13].

### 3.1. In_x_Ga_1−x_N/GaN DQWs

First, we investigated In_x_Ga_1-x_N/GaN DQWs with $L_{qw,A}=3\;MLs$, $L_{qw,B}=2\;MLs$, and $L_{ib}=2\;MLs$, which were grown on conventional GaN substrates. In Figure 2, we show the bulk energy gap, $E_{2Dg}$, and the subband dispersions for four distinct phases occurring in these structures. Figure 2a presents the $E_{2Dg}$ as a function of the In content in the QWs. As in the case of single In_x_Ga_1−x_N/GaN QWs [13,20], we observed the TPT and the NTPT, which were accompanied by the closing of $E_{2Dg}$. In a more detailed analysis, we observed that when the In content of the QWs, x, is below 0.96307, the DQW system is in the NI phase with the usual ordering of sub-bands (see Figure 2b). We would like to note that due to the negative SOI of InN, the highest light-hole (LH) sub-band with the $\Gamma_{7}$ symmetry is above the highest heavy-hole (HH) sub-band with the $\Gamma_{9}$ symmetry [13]. The names of the subbands reflect the dominant contribution of the CB, HH, and LH states around ${\vec{k}}_{⟘}=0$ [19]. As the value of x increases, the energy gap of In_x_Ga_1−x_N alloys decreases toward the bandgap of InN and, more importantly, the built-in electric field in the QWs increases, causing an inversion of the CB and LH sub-bands and the TPT from the NI to the TI (see Figure 2d). The TPT is mediated by the Weyl semimetal (WSM) (see Figure 2c) because the CB and LH sub-bands anticross at ${\vec{k}}_{⟘}=0$ [13]. The amplitude of compressive in-plane strain in the QW layers at the TPT, denoted by $\left|\varepsilon_{xx,qw}^{TPT}\right|$, is about 9.71%. In the TI phase, the $E_{2Dg}^{TI}$ reaches a maximum value of $E_{2Dg,max}^{TI}=0.826\;\mathrm{meV}$. For x values larger than 0.96667, the $E_{2Dg}$ vanishes due to the NTPT from the TI phase to the NTSM, arising from nonlocal overlapping between the sub-bands, as shown in Figure 2e [13,20].

Similar but slightly different results were obtained for In_x_Ga_1−x_N/GaN DQWs with $L_{qw,A}=2\;MLs$, $L_{qw,B}=3\;MLs$, and $L_{ib}=2\;MLs$. Figure 3a depicts the $E_{2Dg}$ for these structures as a function of x. The TPT and the NTPT occurs for x = 0.95467 and x = 0.95812, respectively. The $\left|\varepsilon_{xx,qw}^{TPT}\right|$ is 9.63% whereas the $E_{2Dg,max}^{TI}=0.632\;\mathrm{meV}$. The differences between the results presented in Figure 2a and Figure 3a originate from the fact that wurtzite structures have no center of inversion, and the crystallographic directions [0001] and [000-1] are not equivalent. In Figure 3b, we present the $E_{2Dg}$ for the In_x_Ga_1−x_N/GaN DQWs with $L_{qw,A}=3\;MLs$, $L_{qw,B}=3\;MLs$, and $L_{ib}=2\;MLs$. Although we dealt with structures having identical widths of QWs, the observed phases remain essentially the same because the built-in electric field breaks the mirror symmetry of the DQW potential (see Figure 1). Therefore, the BG phase, which has been observed for symmetric HgTe/CdTe DQWs [29,30], does not appear in In_x_Ga_1−x_N/GaN DQWs with the identical QW widths. Comparing the results shown in Figure 3b with those presented in Figure 2a and Figure 3a, we see that for the DQWs with both wells having three *MLs*, the TI phase is obtained with significantly less In content and, subsequently, less strain. In particular, the TPT and NTPT occur for x = 0.85573 and x = 0.85712, respectively. The $\left|\varepsilon_{xx,qw}^{TPT}\right|$ is 8.71%, which is the advantage of these structures in terms of their epitaxial growth. Unfortunately, we predicted that the $E_{2Dg,max}^{TI}=0.427\;\mathrm{meV}$, which is almost twice as small as the DQWs considered in Figure 2a.

### 3.2. InN/In_y_Ga_1−y_N DQWs

We extended our investigations to InN/In_y_Ga_1−y_N DQWs. We assumed that these structures are pseudomorphically grown on metamorphic In_y_Ga_1−y_N buffer layers or In_y_Ga_1−y_N virtual substrates, which are used in optoelectronic devices [42,43,44,45]. In Figure 4, we present the $E_{2Dg}$ for InN/In_y_Ga_1−y_N DQWs with (a) $L_{qw,A}=3\;MLs$ and $L_{qw,B}=2\;MLs$, (b) $L_{qw,A}=2\;MLs$ and $L_{qw,B}=3\;MLs$, and (c) $L_{qw,A}=3\;MLs$ and $L_{qw,B}=3\;MLs$. The width of the interwell barrier is $L_{ib}=2\;MLs$. In all cases, we see the TPT from the NI to the TI via the WSM and the NTPT from the TI to the NTSM. These phase transitions are driven by an increase in the built-in electric field in QWs, due to a decrease in the In content in the barriers, y. For the structures presented in Figure 4a–c, the TPT occurs at a y equal to 0.09345, 0.10937, and 0.3311, respectively. Therefore, the values of $\left|\varepsilon_{xx,qw}^{TPT}\right|$ are 9.10%, 8.94%, and 6.72%, and they are significantly smaller compared with the results obtained for the corresponding In_x_Ga_1−x_N/GaN DQWs. Moreover, for InN/In_y_Ga_1−y_N DQWs, we obtained higher values of $E_{2Dg,max}^{TI}$, which are equal to 1.066, 0.908, and 1.178 meV, for the structures considered in Figure 4a–c, respectively. Interestingly, in the case of InN/In_y_Ga_1−y_N DQWs, we found that the smallest value of $\left|\varepsilon_{xx,qw}^{TPT}\right|$ and, simultaneously, the largest $E_{2Dg,max}^{TI}$, are for the structures with $L_{qw,A}=3\;MLs$ and $L_{qw,B}=3\;MLs$. Therefore, these structures are the most attractive for experimental observation of the QSHE.

Finally, we studied the effect of increasing $L_{ib}$ on the phase transitions in InN/In_y_Ga_1−y_N DQWs. Figure 5 shows the $E_{2Dg}$ for the structures with (a) $L_{qw,A}=3\;MLs$, $L_{qw,B}=2\;MLs$, and $L_{ib}=3,\;4,\;5\;MLs$; (b) $L_{qw,A}=2\;MLs$, $L_{qw,B}=3\;MLs$, and $L_{ib}=3,\;4,\;5\;MLs$; and (c) $L_{qw,A}=3\;MLs$, $L_{qw,B}=3\;MLs$, and $L_{ib}=3,\;4,\;5,\;6\;MLs$. We see that the In content in the barriers for obtaining the TPT slightly decreases with increasing $L_{ib}$, so the $\left|\varepsilon_{xx,qw}^{TPT}\right|$ increases with increasing $L_{ib}$. More importantly, one can see that both the $E_{2Dg,max}^{TI}$ and the window of the In content for the TI phase, $\Delta y_{TI}$, rapidly decrease with increasing $L_{ib}$. This effect is additionally demonstrated in Figure 6, where the $E_{2Dg,max}^{TI}$ and $\Delta y_{TI}$ are presented as a function of $L_{ib}$. Figure 6a shows that with increasing $L_{ib}$, the $E_{2Dg,max}^{TI}$ decreases at a similar rate for all three series of DQWs. In Figure 6b, we see that the reduction in $\Delta y_{TI}$ with increasing $L_{ib}$ is slower for the structures with both QW widths being three *MLs* than for the structures with QW widths of two and three *MLs*. We also found that for sufficiently large $L_{ib}$, i.e., $L_{ib}=5\;MLs\;$in Figure 5a,b and $L_{ib}=6\;MLs\;$in Figure 5c, the TI phase does not appear. The value of $L_{ib}$ at which the TI phase disappears is one *ML* larger for the series of DQWs with both QW widths being three *MLs* (Figure 5c), because for these structures, the $E_{2Dg,max}^{TI}$ is significantly larger than for the other two series of DQWs (Figure 5a,b), as is clearly seen in Figure 6a. In the cases where the TI phase disappears, we observed two novel phase transitions. First, we found the NTPT, from the NI to the nonlocal normal semimetal (NNSM), having the normal ordering of the CB and LH sub-bands. Then, the TPT occurs from the NNSM to the NTSM via the buried Weyl semimetal phase (BWSM) containing the Weyl points, which are buried in the LH sub-band. In Figure 7, we show the sub-band dispersions for all four phases ((a) NI, (b) NNSM, (c) BWSM, and (d) NTSM) in InN/In_y_Ga_1−y_N DQWs with $L_{qw,A}=2\;MLs$, $L_{qw,B}=3\;MLs$, and $L_{ib}=5\;MLs$, which occur in order of decreasing In content in the barriers. Similar results were obtained for the structures with $L_{qw,A}=3\;MLs$, $L_{qw,B}=2\;MLs$, and $L_{ib}=5\;MLs$, and $L_{qw,A}=3\;MLs$, $L_{qw,B}=3\;MLs$, and $L_{ib}=6\;MLs$. We would like to note that the NTPT from the NI to the NNSM was predicted for HgTe/CdTe QWs at high hydrostatic pressure [46]. To the best of our knowledge, the TPT from the NNSM to the NTSM via the BWSM was not discovered in any 2D structure.

## 4. Conclusions

We studied the phase transitions and the properties of the TI phase in In_x_Ga_1−x_N/GaN and InN/In_y_Ga_1−y_N DQWs, applying a realistic model based on the nonlinear theory of elasticity and piezoelectricity, and the eight-band **k·p** method with relativistic and nonrelativistic linear-wave-vector terms. Despite a rigorous assumption of a negative SOI in InN, we demonstrated that the TI phase can occur in In_x_Ga_1−x_N/GaN and InN/In_y_Ga_1−y_N DQWs when the widths of individual QWs are two and three *MLs*, and three and three *MLs*. In these structures, when the interwell barrier is sufficiently thin, we observed the TPT from the NI to the TI via the WSM, and the NTPT from the TI to the NTSM. We found that in In_x_Ga_1−x_N/GaN DQWs, the $E_{2Dg}^{TI}$ is much smaller for the structures with both QW widths being three *MLs* than when the QW widths are two and three *MLs*, whereas in InN/In_y_Ga_1−y_N DQWs, the opposite was true. For InN/In_y_Ga_1−y_N DQWs with 3 *ML* QWs and the $L_{ib}=2\;MLs$, the $E_{2Dg}^{TI}$ can reach about 1.2 meV. Our calculations also revealed that both the $E_{2Dg}^{TI}$ and the $\Delta y_{TI}$ rapidly decrease with increasing $L_{ib}$. We showed that for structures with $L_{ib}$ above 5 or 6 *MLs*, the TI did not occur. In these structures, we found two novel phase transitions, namely the NTPT from the NI to the NNSM and the TPT from the NNSM to the NTSM, via the BWSM. We hope that these results will stimulate intensive theoretical and experimental studies toward achieving the TI phase in InGaN-based DQWs and will contribute to new applications of these prospective topological nanomaterials. Our work lays the groundwork for future investigations of the phase transitions in other QW systems fabricated from nontopological semiconductors, such as Ge/GaAs, InSb/CdTe, and ZnO/CdO, in which an inversion of the CB and VB sub-bands is achieved by the built-in electric field [47,48,49].

## Figures and Tables

**Figure 1 nanomaterials-12-02418-f001:**
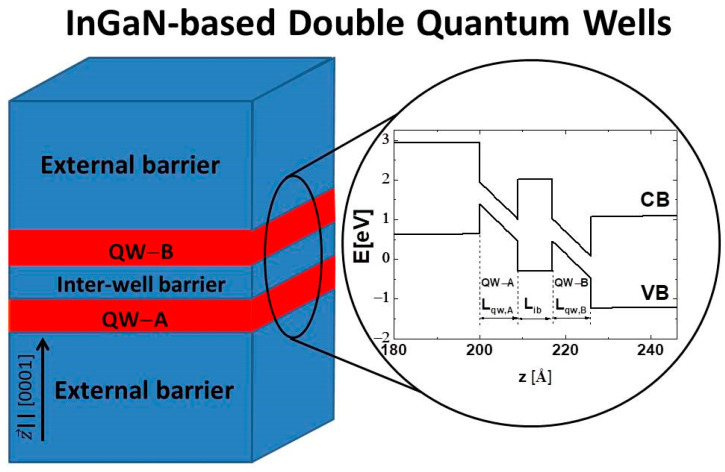
Schematic representation of an InGaN-based DQW heterostructure. On the right side, the CB and VB edge profiles of an exemplary structure containing InN/In_0.3292_Ga_0.6708_N DQWs with $L_{qw,A}=3\;MLs$, $L_{qw,B}=3\;MLs$, and $L_{ib}=3\;MLs$.

**Figure 2 nanomaterials-12-02418-f002:**
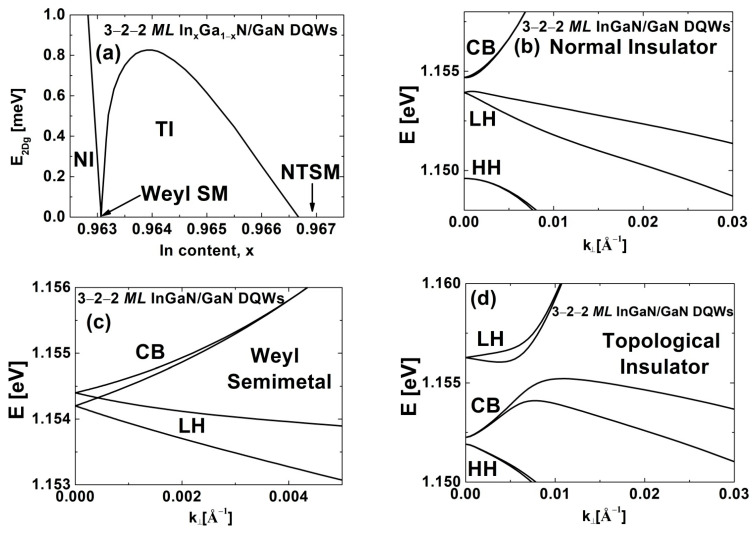
(**a**) The $E_{2Dg}$ for In_x_Ga_1−x_N/GaN DQWs with $L_{qw,A}=3\;MLs$, $L_{qw,B}=2\;MLs$, and $L_{ib}=2\;MLs$ as a function of the In content in the QWs. (**b**–**e**) The sub-band dispersions for (**b**) the NI, (**c**) WSM, (**d**) TI, and (**e**) NTSM occurring in these DQWs.

**Figure 3 nanomaterials-12-02418-f003:**
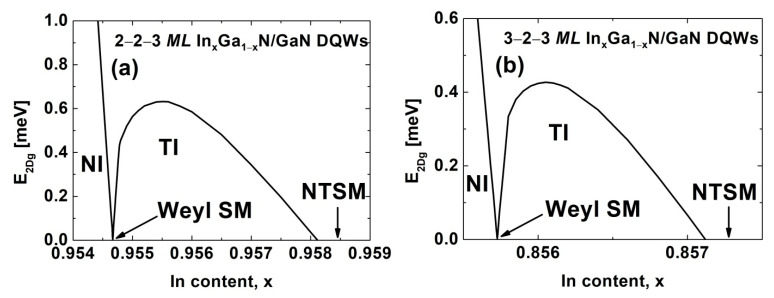
The $E_{2Dg}$ as a function of the In content in the QWs, for In_x_Ga_1−x_N/GaN DQWs with (**a**) $L_{qw,A}=2\;MLs$, $L_{qw,B}=3\;MLs$, and $L_{ib}=2\;MLs$; and (**b**) $L_{qw,A}=3\;MLs$, $L_{qw,B}=3\;MLs$, and $L_{ib}=2\;MLs$.

**Figure 4 nanomaterials-12-02418-f004:**
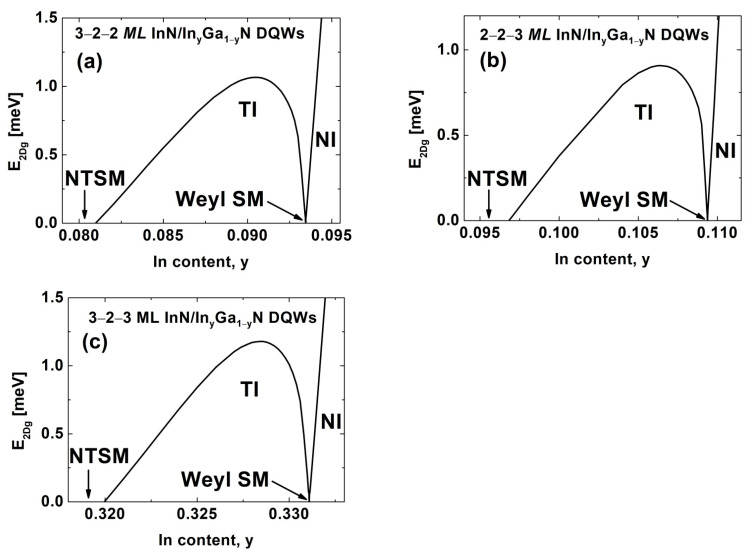
The $E_{2Dg}$ as a function of the In content in the barriers for InN/In_y_Ga_1−y_N DQWs with (**a**) $L_{qw,A}=3\;MLs$, $L_{qw,B}=2\;MLs$, and $L_{ib}=2\;MLs$; (**b**) $L_{qw,A}=2\;MLs$, $L_{qw,B}=3\;MLs$, and $L_{ib}=2\;MLs$; and (**c**) $L_{qw,A}=3\;MLs$, $L_{qw,B}=3\;MLs$, and $L_{ib}=2\;MLs$.

**Figure 5 nanomaterials-12-02418-f005:**
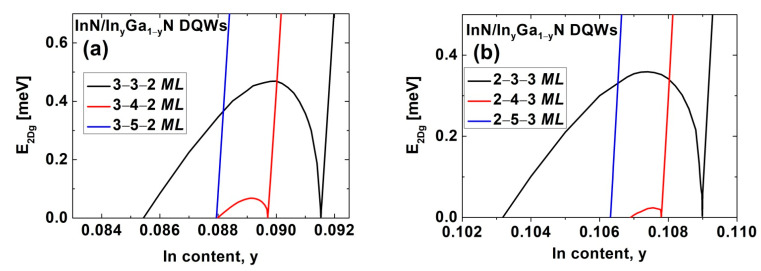
The $E_{2Dg}$ as a function of the In content in the barriers for InN/In_y_Ga_1−y_N DQWs with (**a**) $L_{qw,A}=3\;MLs$, $L_{qw,B}=2\;MLs$, and $L_{ib}=3,\;4,\;5\;MLs$; (**b**) $L_{qw,A}=2\;MLs$, $L_{qw,B}=3\;MLs$, and $L_{ib}=3,\;4,\;5\;MLs$; and (**c**) $L_{qw,A}=3\;MLs$, $L_{qw,B}=3\;MLs$, and $L_{ib}=3,\;4,\;5,6\;MLs$. The results obtained for different values of $L_{ib}$ are marked with different colors.

**Figure 6 nanomaterials-12-02418-f006:**
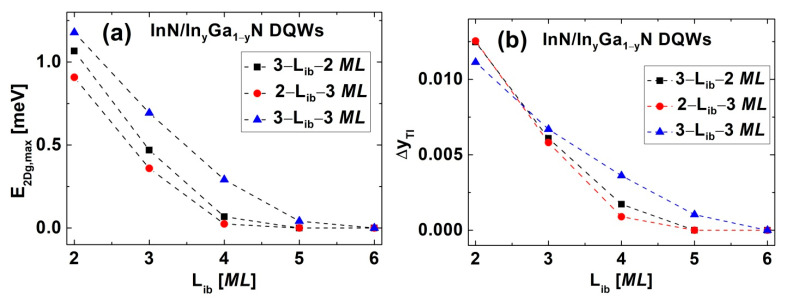
The values of (**a**) $E_{2Dg,max}^{TI}$ and (**b**) $\Delta y_{TI}$ for InN/In_y_Ga_1−y_N DQWs as a function of $L_{ib}$. Squares represent DQWs with $L_{qw,A}=3\;MLs$ and $L_{qw,B}=2\;MLs$, circles represent structures with $L_{qw,A}=2\;MLs$ and $L_{qw,B}=3\;MLs$, and triangles correspond to the results for DQWs with $L_{qw,A}=3\;MLs$ and $L_{qw,B}=3\;MLs$.

**Figure 7 nanomaterials-12-02418-f007:**
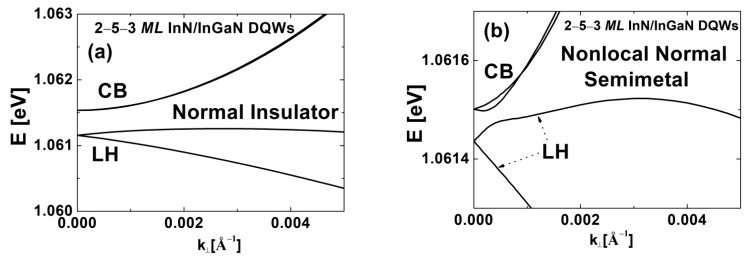
The sub-band dispersions for (**a**) the NI, (**b**) NNSM, (**c**) BWSM, and (**d**) NTSM occurring in InN/In_y_Ga_1−y_N DQWs with $L_{qw,A}=2\;MLs$, $L_{qw,B}=3\;MLs$, and $L_{ib}=5\;MLs$. The phases appear in order of decreasing In content in the barriers.

## Data Availability

The data underlying this article are available from the corresponding author upon reasonable request.

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
