# Peer review of "Polarization-Induced Phase Transitions in Ultra-Thin InGaN-Based Double Quantum Wells"

_nanomaterials, 2022, doi:10.3390/nano12142418_

Round 1

Reviewer 1 Report

This Letter investigate the phase transitions and the properties of the topological insulator in InGaN/GaN and InN/InGaN double quantum wells. The following concerns need to be stressed before publication.

(1)   English expression and grammars need to be checked.

(2)   In Fig. 5, the TI phase disappears at different ??? (5 or 6). Can the author explant the reason for the different?

Some subgraphs are not described in the text. Be sure to cite every figure.

Reviewer 2 Report

In this paper the authors study the phase transition in thin InGaN-based double quantum wells.  They investigated the possibility of topological phase transition in quantum double  wells in InxGa1-xN/GaN and InN/InyGa1-yN by adjusting several parameters such as the depth of quantum wells.  They calculated the bulk energy gap E2Dg and the band dispersions.  The results may be helpful in the search for new topological materials.  I have several comments as follows.

(1) ML (monolayer) may be a unit of depth.  What is the thickness of one layer?

(2) Many contributions such as piezoelectric effect, polarization and k.p terms are included in calculations following Refs. 13 and 31.  Please add discussion on what is essential for the topological transition to occur.

(3) Because there are so many abbreviations, the paper is difficult to read.  There are abbreviations that are not commonly used, for example, SOEC (second-order elastic constant) and TOEC (third-order elastic constant).     

Reviewer 3 Report

The authors investigate topological phase transition in In ga structur. I recommend the publication of the work as it presents a novel study. The manuscript is written well. The results are clearly presented and look sound to me. My only major comment is related to the conclusion of the paper where the results are only summerized without providing an outlook for future study. Also in the abstract thee implication of the work is not presented well. Yet, overall I Vielecke that the manuscript deserve publication.

Round 2

Reviewer 2 Report

The manuscript has been improved following suggestions by the referee.  I think that the paper can be published in  present form.